# HRVMamba: High-Resolution Visual State Space Model for Dense Prediction

## Abstract

Recently, State Space Models (SSMs) with efficient hardware-aware designs, *i.e.*, Mamba, have demonstrated significant potential in computer vision tasks due to their linear computational complexity with respect to token length and their global receptive field. However, Mamba's performance on dense prediction tasks, including human pose estimation and semantic segmentation, has been constrained by three key challenges: insufficient inductive bias, long-range forgetting, and low-resolution output representation. To address these challenges, we introduce the Dynamic Visual State Space (DVSS) block, which utilizes multi-scale convolutional kernels to extract local features across different scales and enhance inductive bias, and employs deformable convolution to mitigate the long-range forgetting problem while enabling adaptive spatial aggregation based on input and task-specific information. By leveraging the multi-resolution parallel design proposed in HRNet (Wang et al., 2020), we introduce High-Resolution Visual State Space Model (HRVMamba) based on the DVSS block, which preserves high-resolution representations throughout the entire process while promoting effective multi-scale feature learning. Extensive experiments highlight HRVMamba's impressive performance on dense prediction tasks, achieving competitive results against existing benchmark models without bells and whistles. We will make the source code publicly accessible.

## 1 Introduction

Convolutional Neural Networks (CNNs)(He et al., 2016; Liu et al., 2022; Zhang et al., 2023; 2024a;b) and Vision Transformers (ViTs)(Yuan et al., 2021; Liu et al., 2021; Shaker et al., 2023; Yun & Ro, 2024) have driven significant progress in tasks like image classification, human pose estimation, and semantic segmentation. While CNNs excel at local feature extraction with linear computational complexity, they lack global context modeling. ViTs, despite capturing global receptive fields via self-attention, face quadratic complexity and lack inductive bias, especially with large inputs. Mamba (Gu & Dao, 2023) introduces the S6 structure, improving the efficiency of State Space Models (SSMs) for long-range feature extraction. By using input-dependent state-space parameters, Mamba enables better context modeling with linear complexity. This led to many follow-up visual Mamba models like ViM (Zhu et al., 2024), VMamba (Liu et al., 2024b), LocalVMamba (Huang et al., 2024), GroupMamba (Shaker et al., 2024), and MambaVision (Hatamizadeh & Kautz, 2024).

However, visual Mamba models have not achieved optimal performance on dense prediction tasks, including human pose estimation (Xu et al., 2024; Zhang et al., 2024d) and semantic segmentation (Wang et al., 2020; Touvron et al., 2021), due to three key challenges. Firstly, like ViT, visual Mamba splits images into sequences of patches (tokens) and employs either a bidirectional (Zhu et al., 2024; Li et al., 2024) or four-way scanning mechanism (Liu et al., 2024b; Huang et al., 2024) to traverse these tokens, constructing a global receptive field. While this approach is effective for handling long sequences, it disrupts the natural 2D spatial dependencies of images and lacks the inductive bias crucial for effective local representation learning. Secondly, Mamba's token processing leads to the decay of the previous hidden state, resulting in long-range forgetting. Consequently, it may lose high-level, task-specific features relevant to the query patch and instead focus on low-level edge features, as shown in Fig. 1, column 2[1]. Lastly, current visual Mamba models usually generate

---

[1]We use the attention activation map visualization method proposed by VMamba (Liu et al., 2024b).

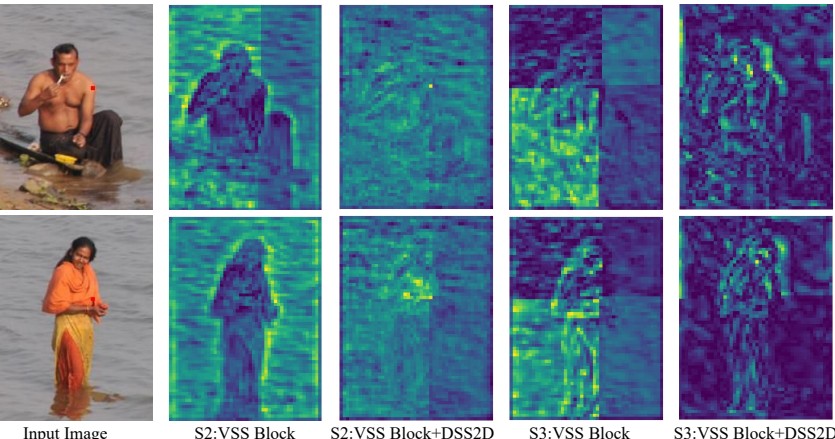

Input Image    S2:VSS Block    S2:VSS Block+DSS2D    S3:VSS Block    S3:VSS Block+DSS2D

Figure 1: **Activation maps of SSM for the query patch** (marked by a red rectangle). We embed the VSS block (Liu et al., 2024b) and VSS+DSS2D (with the SS2D block replaced by our proposed DSS2D block) into the HRVMamba-S. $Si$ represents the $i$-th stage of HRVMamba. We present the activation map of the SSM from the second block of the first module of the first branch in $Si$. In the early stage (S2), DSS2D block focuses on high-level features related to the query patch, while SS2D block targets low-level edge features. In the later stage (S3), DSS2D block highlights human-related details, whereas SS2D block captures more irrelevant background information.

single-scale, low-resolution features, which cause substantial information loss and make it difficult to capture the fine-grained details and multi-scale variations necessary for dense prediction tasks.

To address these limitations, we introduce the Dynamic Visual State Space (DVSS) Block, building on the Visual State Space (VSS) block proposed in VMamba. The DVSS block utilizes multi-scale convolutional kernels to extract local features across different scales, enhancing inductive bias for a range of visual feature scales. Additionally, it integrates the $3 \times 3$ Deformable Convolution v4 (DCNv4)(Xiong et al., 2024), enabling dynamic high-level spatial aggregation based on input and task-specific information. This mitigates Mamba's long-range forgetting issue by enhancing high-level semantic relationships between patches, enabling them to influence one another despite long-range decay, rather than primarily focusing on low-level features. For example, as shown in Fig.1, the left shoulder and chest features near the right shoulder are highlighted (row 1, column 3), the head features connected to the right shoulder are emphasized (row 1, column 5), along with the highlighted features of both hands and the chest (row 2, column 3). We further adopt the multi-resolution parallel design from HRNet (Wang et al., 2020), embedding the DVSS block into parallel multi-resolution branches to construct the High-Resolution Visual State Space Model (HRVMamba). HRVMamba maintains and enhances high-resolution representations, preserving more fine-grained details and modeling multi-scale variations with the multi-resolution branches, making it well-suited for dense prediction tasks.

The contributions of this study are as follows:

- We introduce the DVSS block, which integrates multi-scale convolutional kernels and deformable convolutions to mitigate the lack of inductive bias and long-range forgetting issues in the VSS block.

- We propose HRVMamba, based on the DVSS block, as the first Mamba-based model applied in a multi-resolution branch structure, designed to preserve fine-grained details and capture multi-scale variations specifically for dense prediction tasks.

- HRVMamba demonstrates promising performance in image classification, human pose estimation, and semantic segmentation tasks. Experimental results show that HRVMamba achieves competitive results against existing CNN, ViT, and SSM benchmark models.

## 2    RELATED WORK

**Convolutional Neural Networks (CNNs) and Vision Transformers (ViTs).** CNNs have long been the cornerstone of computer vision, evolving from early models like AlexNet (Xiong et al., 2024) and ResNet (He et al., 2016) to more recent architectures such as ConvNeXt (Liu et al., 2022),

SCGNet (Zhang et al., 2023), FlashInternImage (Xiong et al., 2024), and FMGNet (Zhang et al., 2024b). These models excel in local feature extraction, achieving remarkable performance across tasks such as image classification, semantic segmentation, and human pose estimation. ViTs introduce self-attention mechanisms from natural language processing (NLP), segmenting images into patches to capture global dependencies, forming the foundation of Large Vision-Language Models (Zhang et al., 2024c; Ying et al., 2024; Liu et al., 2024a). Various methods, including DeiT's (Touvron et al., 2021) distillation strategies, Swin Transformer's (Liu et al., 2021) hierarchical structures, and SwiftFormer's (Shaker et al., 2023) efficient attention mechanisms, have been developed to broaden the adoption of ViTs in vision tasks. Recently, hybrid architectures (Yun & Ro, 2024; Ma et al., 2024) that combine the strengths of CNNs and Transformers have gained attention. These models leverage the inductive biases of CNNs for local feature extraction while incorporating the global attention capabilities of ViTs, marking a significant direction in backbone network research.

**State Space Models (SSMs).** SSMs are a mathematical framework for modeling dynamic systems with linear computational complexity, making them efficient for long sequences. Optimizations in models like S4 (Gu et al., 2021), S5 (Smith et al., 2022), and H3 (Fu et al., 2022) have enhanced SSMs performance through structure optimization, parallel scanning and hardware improvements. Mamba (Gu & Dao, 2023) introduces input-specific parameterization and parallel scanning (S6), positioning SSMs as a compelling alternative to Transformers. Since then, SSMs have been widely adopted in vision tasks, with S4ND (Nguyen et al., 2022) being one of the first to process visual data as continuous signals. Building on Mamba, models like ViM (Zhu et al., 2024) and VMamba (Liu et al., 2024b) address the direction-sensitivity of Mamba with bidirectional or four-way scanning. LocalVMamba (Huang et al., 2024) captures local details with windowed scanning, while PlainMamba (Yang et al., 2024) refines 2D scanning for sequential processing. MambaVision (Hatamizadeh & Kautz, 2024) integrates SSMs with Transformers, and GroupMamba (Shaker et al., 2024) improves training stability with a distillation-based approach. However, these visual Mamba models often produce single-scale, low-resolution features, limiting their ability to capture the fine-grained details and multi-scale variations required for dense prediction tasks.

**High-Resolution Networks for Dense Prediction.** The High-Resolution network was first introduced in HRNet (Wang et al., 2020), demonstrating strong performance in tasks such as human pose estimation and semantic segmentation. Its multi-resolution parallel branches combine information at various scales by exchanging multi-resolution features through a multi-scale fusion module. Building on this, HRFormer (Yuan et al., 2021) integrates the local-window self-attention mechanism (Liu et al., 2021) with the high-resolution structure, achieving excellent results in dense prediction tasks. Further advancements, including Lite-HRNet (Yu et al., 2021), Dite-HRNet (Li et al., 2022), and HF-HRNet (Zhang et al., 2024a), introduce lightweight CNNs through techniques such as depthwise and dynamic convolutions, enabling high-resolution networks to be more efficiently deployed on mobile devices. However, whether Mamba can perform optimally within high-resolution structures and how to mitigate Mamba's lack of inductive bias and long-range forgetting in dense prediction tasks remain open areas for further investigation.

## 3 PRELIMINARIES

**State Space Models (SSMs)** map a 1D function or sequence $x(t) \in \mathbb{R}$ to output sequence $y(t) \in \mathbb{R}$ though a hidden state $\boldsymbol{h} \in \mathbb{R}^N$ based on continuous linear time-invariant (LTI) systems. To integrate deep models and adapt to real-world data, discretization must be applied to convert the continuous differential equations of SSMs into discrete functions using the zero-order hold method. Specifically, with a discrete-time step $\Delta \in \mathbb{R}$, SSMs are discretized as follows:

$$\boldsymbol{h}_t = \bar{\mathbf{A}}\boldsymbol{h}_{t-1} + \bar{\mathbf{B}}x_t, \tag{1}$$

$$y_t = \mathbf{C}^\top \boldsymbol{h}_t, \tag{2}$$

where $x_t = x(\Delta t)$, $\mathbf{A} \in \mathbb{R}^{N \times N}$ is the system's evolution matrix, and $\mathbf{B} \in \mathbb{R}^{N \times 1}$ and $\mathbf{C} \in \mathbb{R}^{N \times 1}$ are the projection matrices. $\bar{\mathbf{A}} = \exp(\Delta \mathbf{A})$, $\quad \bar{\mathbf{B}} = (\Delta \mathbf{A})^{-1}(\exp(\Delta \mathbf{A}) - \mathbf{I}) \cdot \Delta \mathbf{B} \approx \Delta \mathbf{B}$.

**Selective State Space Models (S6)** are introduced in Mamba (Gu & Dao, 2023) to improve the extraction of strong contextual information. S6 allows $\mathbf{B}$, $\mathbf{C}$, and $\Delta$ to vary as functions of the input $x_t$, whereas in S4 (Gu et al., 2021), $\mathbf{A}$, $\mathbf{B}$, $\mathbf{C}$, and $\Delta$ are input-independent, which limits the model's

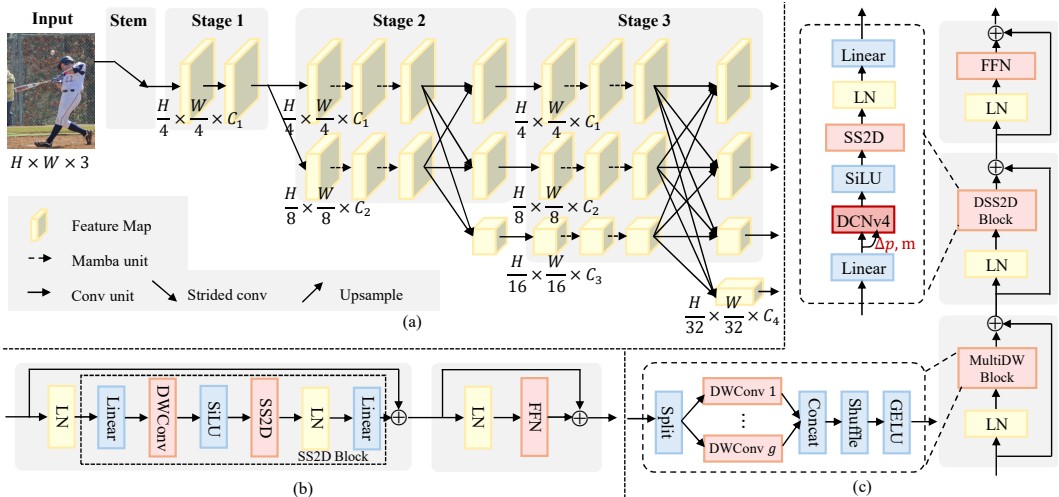

Figure 2: **(a) Overall architecture of HRVMamba.** HRVMamba has four stages, but for demonstration purposes, we only show three. $H$ and $W$ represent the height and width of the image, while $C_i$ denotes the number of channels in the $i$-th branch. **(b) VSS block proposed in VMamba (Liu et al., 2024b). (c) Dynamic Visual State Space block.** LN, Linear, DWConv and SS2D represent LayerNorm, Linear Layer, depthwise convolution, and 2D-Selective-Scan SSM (Liu et al., 2024b).

ability to extract crucial information from the input sequence. Formally, given an input sequence $\boldsymbol{x} \in \mathbb{R}^{B \times L \times C}$, where $B$, $L$, and $C$ represent the batch size, sequence length, and feature dimension, respectively, the input-dependent parameters $\mathbf{B}$, $\mathbf{C}$, and $\Delta$ are computed as follows:

$$\mathbf{B} = \texttt{Linear}(\boldsymbol{x}) \in \mathbb{R}^{B \times L \times N}, \tag{3}$$

$$\mathbf{C} = \texttt{Linear}(\boldsymbol{x}) \in \mathbb{R}^{B \times L \times N}, \tag{4}$$

$$\Delta = \texttt{SoftPlus}(\tilde{\Delta} + \texttt{Linear}(\boldsymbol{x})) \in \mathbb{R}^{B \times L \times C}, \tag{5}$$

where $\tilde{\Delta} \in \mathbb{R}^{B \times L \times C}$ is a learnable parameter, and $\mathbf{A} \in \mathbb{R}^{L \times C}$ is the same parameter as in S4.

## 4 HIGH-RESOLUTION VISUAL STATE SPACE MODEL

### 4.1 MULTI-RESOLUTION PARALLEL VMAMBA

Current visual Mamba models typically generate single-scale, low-resolution features, resulting in significant information loss and difficulty in capturing the fine details and multi-scale variations needed for dense prediction tasks. To address this, we adopt HRNet's multi-resolution parallel design (Wang et al., 2020), using parallel branches to develop the High-Resolution Visual State Space Model (HRVMamba). We illustrate the overall architecture of HRVMamba in Fig. 3 (a). Given an input image $\mathbf{X} \in \mathbb{R}^{H \times W \times 3}$, HRVMamba begins with a downsampling stem using two $3 \times 3$ convolutions with a stride of 2, reducing the feature resolution to $\frac{H}{4} \times \frac{W}{4}$. It then progresses through four stages, where the streams in later stages include the previous stage's resolutions and an additional lower one, with the final four branches having feature dimensions of $\frac{H}{4} \times \frac{W}{4} \times C_1$, $\frac{H}{8} \times \frac{W}{8} \times C_2$, $\frac{H}{16} \times \frac{W}{16} \times C_3$, and $\frac{H}{32} \times \frac{W}{32} \times C_4$. Previous studies (Yun & Ro, 2024; Ma et al., 2024) have shown that convolution operations on larger feature maps in early stages are more effective for visual feature extraction, so we adopt the BottleNeck structure in the first stage like HRNet. The remaining stages use our proposed Dynamic Visual State Space (DVSS) block (Fig. 3 (c)) as the basic unit. The multi-scale fusion method, following HRNet, incorporates a series of upsampling and downsampling blocks to merge features from different parallel branches.

### 4.2 DYNAMIC VISUAL STATE SPACE (DVSS) BLOCK

We introduce the DVSS Block, building on the VSS block (Fig.3 (b)) of VMamba(Liu et al., 2024b). As illustrated in Fig. 3 (c), the DVSS Block incorporates the 2D-Selective-Scan with Deformable Convolution (DSS2D) block, the Multi-scale Depthwise (MultiDW) Block, and a Feed-Forward Network (FFN) as feature extraction units.

Table 1: **The architecture configuration of HRVMamba.** MDW, and DSS2D represent the MultiDW block and DSS2D block respectively. $(M_1, M_2, M_3, M_4)$: the number of modules, $(B_1, B_2, B_3, B_4)$: the number of blocks, $(S_1, S_2, S_3, S_4)$: the SSM expansion ratios, $(R_1, R_2, R_3, R_4)$: the MLP expansion ratios.

| Res. | Stage 1 | Stage 2 | Stage 3 | Stage 4 |
|---|---|---|---|---|
| $4\times$ | $\begin{bmatrix} 1\times1,64 \\ 3\times3,64 \\ 1\times1,256 \end{bmatrix} \times B_1 \times M_1$ | $\begin{bmatrix} \text{MDW} \\ \text{DSS2D},S_1 \\ \text{FFN},R_1 \end{bmatrix} B_2\times M_2$ | $\begin{bmatrix} \text{MDW} \\ \text{DSS2D},S_1 \\ \text{FFN},R_1 \end{bmatrix} B_3\times M_3$ | $\begin{bmatrix} \text{MDW} \\ \text{DSS2D},S_1 \\ \text{FFN},R_1 \end{bmatrix} B_4\times M_4$ |
| $8\times$ | | $\begin{bmatrix} \text{MDW} \\ \text{DSS2D},S_2 \\ \text{FFN},R_2 \end{bmatrix} B_2\times M_2$ | $\begin{bmatrix} \text{MDW} \\ \text{DSS2D},S_2 \\ \text{FFN},R_2 \end{bmatrix} B_3\times M_3$ | $\begin{bmatrix} \text{MDW} \\ \text{DSS2D},S_2 \\ \text{FFN},R_2 \end{bmatrix} B_4\times M_4$ |
| $16\times$ | | | $\begin{bmatrix} \text{MDW} \\ \text{DSS2D},S_3 \\ \text{FFN},R_3 \end{bmatrix} B_3\times M_3$ | $\begin{bmatrix} \text{MDW} \\ \text{DSS2D},S_3 \\ \text{FFN},R_3 \end{bmatrix} B_4\times M_4$ |
| $32\times$ | | | | $\begin{bmatrix} \text{MDW} \\ \text{DSS2D},S_4 \\ \text{FFN},R_4 \end{bmatrix} B_4\times M_4$ |

Table 2: **Architecture details of HRVMamba variants.** HRVMamba-S and HRVMamba-B represent the small and base HRVMamba model, respectively.

| Model | #channels $(C_1, C_2, C_3, C_4)$ | #blocks $(B_1, B_2, B_3, B_4)$ | #modules $(M_1, M_2, M_3, M_4)$ | #SSM ratio $(S_1, S_2, S_3, S_4)$ | #MLP ratio $(R_1, R_2, R_3, R_4)$ |
|---|---|---|---|---|---|
| HRVMamba-S | $(32, 64, 128, 256)$ | $(2, 2, 2, 2)$ | $(1, 1, 4, 2)$ | $(2, 2, 2, 2)$ | $(2, 2, 2, 2)$ |
| HRVMamba-B | $(80, 160, 320, 640)$ | $(2, 2, 2, 2)$ | $(1, 1, 4, 2)$ | $(2, 2, 2, 2)$ | $(2, 2, 2, 2)$ |

**2D-Selective-Scan with Deformable Convolution (DSS2D) Block.** As outlined in previous study (Shi et al., 2024), the contribution of the $m$-th token to the $n$-th token ($m < n$, where $m$ and $n$ are token indices) in SSM for an input sequence $\boldsymbol{x} \in \mathbb{R}^{1\times L\times C}$ can be represented as:

$$\mathbf{C}_n^\top \prod_{i=m}^{n} \bar{\mathbf{A}}_i \bar{\mathbf{B}}_m = \mathbf{C}_n^\top \bar{\mathbf{A}}_{(m\to n)} \bar{\mathbf{B}}_m, \text{ where } \bar{\mathbf{A}}_{(m\to n)} = exp\sum_{i=m}^{n} \Delta_i \mathbf{A}. \tag{6}$$

Typically, the learned $\Delta_i \mathbf{A}$ is negative, causing the exponential factor $\bar{\mathbf{A}}_{(m\to n)}$ in Eq. 6 to decrease significantly as the sequence distance increases. This leads to a consistent weakening of the previous hidden state with each new token, resulting in the **long-range forgetting issue**. Consequently, SSM may lose high-level, task-specific features relevant to the query patch and instead focus on low-level edge features, as illustrated in Fig. 1, column 2.

To mitigate the long-range forgetting issue, we replace the Depthwise convolution in the SS2D block with a $3 \times 3$ Deformable Convolution v4 (DCNv4)(Xiong et al., 2024) and build the DSS2D block. Given an input $\mathbf{X} \in \mathbb{R}^{H\times W\times C}$, the DCNv4 operation with $K$ ($K = 9$ for $3 \times 3$ DCNv4) points is defined for each reference point $p_0$ as:

$$\mathbf{Y}_g = \sum_{k=1}^{K} \mathbf{m}_{gk} \mathbf{X}_g(p_0 + p_k + \Delta p_{gk}), \tag{7}$$

$$\mathbf{Y} = \text{Concat}([\mathbf{Y}_1, \mathbf{Y}_2, ..., \mathbf{Y}_G], \text{axis=-1}), \tag{8}$$

where $G$ represents the number of spatial aggregation groups, set to 4. The scalar $\mathbf{m}_{gk}$ represents the dynamic modulation weight of the $k$-th sampling point in the $g$-th group. $p_k$ is the $k$-th grid sampling location $\{(-1, -1), (-1, 0), ..., (0, +1), ..., (+1, +1)\}$, and $\Delta p_{gk}$ is its dynamic offset.

On the one hand, DCNv4 mitigates Mamba's long-range forgetting issue by enhancing high-level semantic relationships between patches like the attention mechanism, rather than primarily focusing on low-level features. This may help ensure that high-level semantic relationships between patches still influence distant tokens despite long-range decay. On the other hand, DCNv4 improves long-range feature modeling with much higher computational efficiency compared to larger convolutional kernels and self-attention mechanisms.

**Multi-scale Depthwise (MultiDW) Block.** Visual Mamba processes images as token sequences using bidirectional or four-way scanning to establish a global receptive field. However, this approach

Table 3: **Comparison on the COCO pose estimation `val` set.** "*Trans.*" means transformer architecture. $-$ means the numbers are not provided in the original paper. $\dagger$ marks a model that is not pretrained, while $\ddagger$ signifies that the backbone uses the classic decoder from ViTPose (Xu et al., 2024). The #param. and FLOPs of HRFormer (Yuan et al., 2021) are based on the implementation from MMPOSE (Contributors, 2020).

| Arch. | Method | input size | #param. | FLOPs | AP | $AP^{50}$ | $AP^{75}$ | $AP^M$ | $AP^L$ | AR |
|---|---|---|---|---|---|---|---|---|---|---|
| *CNN* | HRNet-W48 (Wang et al., 2020) | $256 \times 192$ | 63.6M | 14.6G | 75.1 | 90.6 | 82.2 | 71.5 | 81.8 | 80.4 |
| | FlashInternImage-B$^{\ddagger}$ (Xiong et al., 2024) | $256 \times 192$ | 100.7M | 17.0G | 74.1 | 90.6 | 82.0 | 70.3 | 80.4 | 79.3 |
| *Trans.* | PRTR (Li et al., 2021a) | $512 \times 384$ | 57.2M | 37.8G | 73.3 | 89.2 | 79.9 | 69.0 | 80.9 | 80.2 |
| | TransPose-H-A6 (Yang et al., 2021) | $256 \times 192$ | 17.5M | 21.8G | 75.8 | $-$ | $-$ | $-$ | $-$ | 80.8 |
| | TokenPose-L/D24 (Li et al., 2021b) | $256 \times 192$ | 27.5M | 11.0G | 75.8 | 90.3 | 82.5 | 72.3 | 82.7 | 80.9 |
| | HRFormer-S (Yuan et al., 2021) | $256 \times 192$ | 7.7M | 3.3G | 74.0 | 90.2 | 81.2 | 70.4 | 80.7 | 79.4 |
| | HRFormer-B (Yuan et al., 2021) | $256 \times 192$ | 43.2M | 14.1G | 75.6 | 90.8 | 82.8 | 71.7 | 82.6 | 80.8 |
| | Swin-B$^{\ddagger}$ (Liu et al., 2021) | $256 \times 192$ | 94.0M | 19.0G | 73.7 | 90.5 | 82.0 | 70.2 | 80.4 | 79.3 |
| | PVTv2-B2$^{\ddagger}$ (Wang et al., 2022) | $256 \times 192$ | 29.1M | 5.1G | 73.7 | 90.5 | 81.2 | 70.0 | 80.6 | 79.1 |
| | ViTPose-B (Xu et al., 2024) | $256 \times 192$ | 90.0M | 17.9G | 75.8 | 90.7 | 83.2 | 68.7 | 78.4 | 81.1 |
| | HRFormer-S (Yuan et al., 2021) | $384 \times 288$ | 7.7M | 7.3G | 75.6 | 90.3 | 82.2 | 71.6 | 82.5 | 80.7 |
| | HRFormer-B (Yuan et al., 2021) | $384 \times 288$ | 43.2M | 30.9G | 77.2 | 91.0 | 83.6 | 73.2 | 84.2 | 82.0 |
| | HRFormer-B$^{\dagger}$ (Yuan et al., 2021) | $384 \times 288$ | 43.2M | 30.9G | 77.0 | 90.8 | 83.3 | 73.2 | 80.7 | 81.8 |
| *SSM* | Vim-S$^{\ddagger}$ (Zhu et al., 2024) | $256 \times 192$ | 28.0M | 6.1G | 69.8 | 89.2 | 78.2 | 67.2 | 75.5 | 76.0 |
| | VMamba-T$^{\ddagger}$ (Liu et al., 2024b) | $256 \times 192$ | 34.7M | 6.0G | 74.4 | 90.4 | 82.3 | 70.8 | 81.0 | 79.6 |
| | VMamba-B$^{\ddagger}$ (Liu et al., 2024b) | $256 \times 192$ | 93.8M | 16.3G | 74.8 | 90.7 | 82.1 | 71.2 | 81.5 | 80.1 |
| | LocalVMamba-S$^{\ddagger}$ (Huang et al., 2024) | $256 \times 192$ | 54.2M | 14.1G | 74.1 | 90.4 | 81.8 | 70.9 | 80.4 | 79.9 |
| | MV-B$^{\ddagger}$ (Hatamizadeh & Kautz, 2024)[2] | $256 \times 192$ | 102.9M | 24.6G | 73.4 | 90.1 | 80.9 | 69.7 | 80.2 | 78.9 |
| | GroupMamba-B$^{\ddagger}$ (Shaker et al., 2024) | $256 \times 192$ | 57.7M | 15.0G | 73.2 | 90.3 | 81.1 | 69.8 | 79.8 | 78.7 |
| *SSM (Ours)* | HRVMamba-S | $256 \times 192$ | 8.0M | 3.3G | 74.6 | 90.5 | 81.7 | 71.1 | 81.0 | 79.9 |
| | HRVMamba-B$^{\dagger}$ | $256 \times 192$ | 47.1M | 14.2G | 76.4 | 90.9 | 83.5 | 73.1 | 82.9 | 81.4 |
| | HRVMamba-S | $384 \times 288$ | 8.0M | 7.4G | 76.4 | 90.9 | 83.3 | 72.7 | 83.0 | 81.3 |
| | HRVMamba-B$^{\dagger}$ | $384 \times 288$ | 47.1M | 32.0G | **77.6** | **91.1** | **84.2** | **74.0** | **84.2** | **82.4** |

disrupts 2D spatial relationships and lacks the inductive bias needed for local features. To address this, we introduce the MultiDW block, which employs multi-scale convolutional kernels to capture local features at various scales, thereby enhancing the inductive bias for features of different scales. Specifically, as shown in Eqs. 9, 10, and 11, the input features $\mathbf{X}$ are first divided into $G$ groups along the channel dimension, where $G$ is set to 4. The $g$-th group undergoes a depthwise convolution with a kernel size of $K_g = 2g + 1$. The resulting features are then concatenated, followed by a shuffle operation to promote feature interaction across the different groups.

$$[\mathbf{X}_1, \mathbf{X}_2, ..., \mathbf{X}_G] = \text{Split}(\mathbf{X}, \text{axis=-1}), \tag{9}$$

$$\mathbf{Y}_g = \text{DWConv } g(K_g \times K_g)(\mathbf{X}_g), \tag{10}$$

$$\mathbf{Y} = \text{GELU}(\text{Shuffle}(\text{Concat}([\mathbf{Y}_1, \mathbf{Y}_2, ..., \mathbf{Y}_G], \text{axis=-1})), \tag{11}$$

where GELU represents the activation function.

### 4.3 HRVMAMBA ARCHITECTURE INSTANTIATION.

We illustrate the architecture configurations of HRVMamba in Table 1. In the $i$-th stage, $B_i$, $S_i$, $R_i$, and $M_i$ represent the number of blocks, the SSM expansion ratio, the MLP expansion ratio, and the number of modules, respectively. Furthermore, we designed two scales of HRVMamba, namely HRVMamba-S and HRVMamba-B. Table 2 presents the details of the HRVMamba variants.

## 5 EXPERIMENTS

### 5.1 HUMAN POSE ESTIMATION

**Training setting.** We evaluate HRVMamba on COCO dataset (Lin et al., 2014) for human pose estimation, which comprises over 200,000 images and 250,000 labeled person instances with 17 keypoints. Our experiments train on the COCO `train` 2017 dataset, which includes 57,000 images and 150,000 person instances. The performance of our model is assessed on the `val` 2017 and

Table 4: **Comparison on the COCO pose estimation `test-dev` set.** [†] marks a model that is not pretrained, while [‡] signifies that the backbone uses the classic decoder from ViTPose.

| Method | input size | #param. | FLOPs | AP | $AP^{50}$ | $AP^{75}$ | $AP^{M}$ | $AP^{L}$ | AR |
|---|---|---|---|---|---|---|---|---|---|
| HRNet-W48 (Wang et al., 2020) | $384 \times 288$ | 63.6M | 32.9G | 75.5 | 92.5 | 83.3 | 71.9 | 81.5 | 80.5 |
| PRTR (Li et al., 2021a) | $512 \times 384$ | 57.2M | 37.8G | 72.1 | 90.4 | 79.6 | 68.1 | 79.0 | 79.4 |
| TransPose-H-A6 (Yang et al., 2021) | $256 \times 192$ | 17.5M | 21.8G | 75.0 | 92.2 | 82.3 | 71.3 | 81.1 | − |
| TokenPose-L/D24 (Li et al., 2021b) | $384 \times 288$ | 29.8M | 22.1G | 75.9 | 92.3 | 83.4 | 72.2 | 82.1 | 80.8 |
| HRFormer-S (Yuan et al., 2021) | $384 \times 288$ | 7.7M | 7.3G | 74.5 | 92.3 | 82.1 | 70.7 | 80.6 | 79.8 |
| HRFormer-B (Yuan et al., 2021) | $384 \times 288$ | 43.2M | 30.9G | 76.2 | 92.7 | 83.8 | 72.5 | 82.3 | 81.2 |
| HRFormer-B[†] (Yuan et al., 2021) | $384 \times 288$ | 43.2M | 30.9G | 76.0 | 92.6 | 83.6 | 72.9 | 810.5 | 81.0 |
| Swin-L[‡] (Liu et al., 2021) | $384 \times 288$ | 207.9M | 88.2G | 75.4 | 92.6 | 83.3 | 72.0 | 80.9 | 80.5 |
| ViTPose-B (Xu et al., 2024) | $256 \times 192$ | 90.0M | 17.9G | 75.1 | 92.5 | 83.1 | 72.0 | 80.7 | 80.3 |
| VMamba-B[‡] (Liu et al., 2024b) | $384 \times 288$ | 93.8M | 36.6G | 75.3 | 92.7 | 83.3 | 72.0 | 80.9 | 80.3 |
| HRVMamba-S | $384 \times 288$ | 8.0M | 7.4G | 75.3 | 92.5 | 83.1 | 72.1 | 80.9 | 80.3 |
| HRVMamba-B[†] | $384 \times 288$ | 47.1M | 32.0G | **76.5** | 92.6 | 84.2 | 73.5 | 81.8 | **81.4** |

Table 5: **Performance comparison for semantic segmentation.** We report the mIoUs on Cityscapes `val` and PASCAL-Context `test`. 'SS' and 'MS' denote evaluations performed at single-scale and multi-scale levels, respectively.

| method | #param. | Cityscapes | | PASCAL-Context |
|---|---|---|---|---|
| | | mIoU (SS) | mIoU (MS) | mIoU |
| Swin-B (Liu et al., 2021) | 121M | 66.6 | 67.4 | 36.0 |
| ConvNeXt-B (Liu et al., 2022) | 122M | 71.5 | 71.9 | 39.5 |
| VMamba-B (Liu et al., 2024b) | 122M | 71.8 | 72.0 | 40.7 |
| LocalVMamba-S (Huang et al., 2024) | 81M | 76.3 | 77.0 | 12.2 |
| GroupMamba-B (Shaker et al., 2024) | 83M | 71.5 | 72.0 | 40.3 |
| MV-B (Hatamizadeh & Kautz, 2024) | 130M | 73.6 | 74.5 | 41.3 |
| HRFormer-B (Yuan et al., 2021) | 75M | 77.3 | 77.7 | 42.6 |
| HRVMamba-B | 79M | **79.4** | **80.2** | **43.5** |

`test-dev` 2017 sets, comprising 5,000 and 20,000 images, respectively. For training and evaluation, we follow the implementation of MMPOSE (Contributors, 2020). The batch size is set to 256, and the AdamW optimizer is used, configured with a learning rate of 5e-4, betas of (0.9, 0.999), and a weight decay of 0.01. For HRVMamba-B, no pretraining techniques are employed, whereas, for HRVMamba-S, we apply ImageNet (Deng et al., 2009) pretraining like HRFormer.

**Results.** Table 3 presents the results on the COCO `val` dataset. HRVMamba consistently outperforms other CNN models, ViT models, and recent state-of-the-art (SOTA) SSM methods. With an input size of $256 \times 192$, HRVMamba-S achieves 74.6 AP, exceeding FlashInternImage-B (74.1 AP) while using only one-fifth of the FLOPs. HRVMamba-B achieves 76.4 AP, surpassing SOTA SSM methods like Vim-S, VMamba-B, MambaVision-B, and GroupMamba-B. At similar computational complexity, HRVMamba-B improves by 3.2 AP and 2.7 AR over GroupMamba-B. Additionally, HRVMamba-B outperforms ViTPose-B by 0.6 AP and 0.3 AR, with 50% fewer parameters and 20% fewer FLOPs. With an input size of $384 \times 288$, HRVMamba-S achieves a 0.8 AP gain over HRFormer-S; HRVMamba-B gains 0.6 AP over HRFormer-B without pretraining on ImageNet.

We also provide comparisons on the COCO `test-dev` set in Table 4. Our HRVMamba-S achieves an AP of 75.3, outperforming ViTPose-B by 0.2 while using only one-eleventh of its parameters. It matches the performance of VMamba-B, but with just one-fifth of the FLOPs. Furthermore, HRVMamba-B surpasses HRFormer-B by 0.5 in AP and 0.4 in AR without pretraining on ImageNet, and achieves SOTA performance.

## 5.2 SEMANTIC SEGMENTATION

We adopt UPerNet (Xiao et al., 2018) as the foundational framework for all the models tested. All models are pretrained on the ImageNet-1K dataset (Deng et al., 2009).

**Cityscapes** dataset (Cordts et al., 2016) is designed for urban scene understanding with 19 classes used for semantic segmentation. The finely annotated 5K images are split into 2,975 `train`, 500 `val`, and 1,525 `test` images. We set the initial learning rate to $1e^{-4}$, weight decay to 0.01, crop

Table 6: **Comparison with the state-of-the-art on ImageNet.** "iso.", "hie.", "hig." represent isotropic architecture without downsampling layers, hierarchical architecture, high-resolution architecture, respectively.

| Type | Arch. | Model | Input | #Param (M) | FLOPs (G) | Top-1 Acc |
|------|-------|-------|-------|-----------|-----------|-----------|
| iso. | CNN | ConvNeXt-S (Liu et al., 2022) | $224^2$ | 22 | 4.3 | 79.7 |
|      |      | ConvNeXt-B (Liu et al., 2022) | $224^2$ | 87 | 16.9 | 82.0 |
|      | Trans. | DeiT-S (Touvron et al., 2021) | $224^2$ | 22 | 4.6 | 79.8 |
|      |      | DeiT-B (Touvron et al., 2021) | $224^2$ | 87 | 17.6 | 81.8 |
|      | SSM | S4ND-ViT-B (Nguyen et al., 2022) | $224^2$ | 89 | - | 80.4 |
|      |     | Vim-Ti (Zhu et al., 2024) | $224^2$ | 7 | 1.1 | 76.9 |
|      |     | Vim-S (Zhu et al., 2024) | $224^2$ | 26 | 4.3 | 80.5 |
|      |     | VideoMamba–S (Li et al., 2024) | $448^2$ | 26 | 16.9 | **83.2** |
|      |     | PlainMamba-L3 (Yang et al., 2024) | $224^2$ | 50 | 14.4 | 82.3 |
| hie. | CNN | ConvNeXt-T (Liu et al., 2022) | $224^2$ | 29 | 4.5 | 82.1 |
|      |     | ConvNeXt-B (Liu et al., 2022) | $224^2$ | 89 | 15.4 | 83.8 |
|      | Trans. | Swin-T (Liu et al., 2021) | $224^2$ | 28 | 4.5 | 81.3 |
|      |      | Swin-B (Liu et al., 2021) | $224^2$ | 88 | 15.4 | 83.5 |
|      | SSM | VMamba-B (Liu et al., 2024b) | $224^2$ | 89 | 15.4 | 83.9 |
|      |     | LocalVMamba-S (Huang et al., 2024) | $224^2$ | 50 | 11.4 | 83.7 |
|      |     | ViM2-B (Behrouz et al., 2024) | $224^2$ | 43 | - | 83.7 |
|      |     | MV-B (Hatamizadeh & Kautz, 2024) | $224^2$ | 50 | 15.0 | **84.2** |
| hig. | Trans. | HRFormer-S (Yuan et al., 2021) | $256^2$ | 20 | 5.9 | 80.7 |
|      |      | HRFormer-B (Yuan et al., 2021) | $224^2$ | 57 | 14.2 | 83.2 |
|      | SSM | HRVMamba-S | $256^2$ | 20 | 5.8 | 81.3 |
|      |     | HRVMamba-B | $224^2$ | 61 | 15.8 | **83.7** |

size to 1024×512, batch size to 16, and 80K training iterations. As shown in Table 5, HRVMamba-B outperforms HRFormer-B by 2.1 mIoU in single-scale and 2.5 mIoU in multi-scale tests. It also surpasses SSM models like GroupMamba-B and MambaVision-B with fewer parameters.

**PASCAL-Context** dataset (Mottaghi et al., 2014) includes 59 semantic classes and 1 background class, with 4,998 `train` images and 5,105 `test` images. We set the initial learning rate to $1e^{-4}$, weight decay to 0.01, crop size to 480×480, batch size to 16, and 80K training iterations. As shown in Table 5, HRVMamba-B achieves an improvement of 0.9 mIoU and 2.2 mIoU over HRFormer-B and MambaVision-B, respectively. Notably, LocalVMamba-S performs particularly poorly in tests with variable input sizes, achieving only 12.2 mIoU.

## 5.3 IMAGENET CLASSIFICATION

**Training setting.** We conduct comparisons on the ImageNet-1K dataset (Deng et al., 2009), which comprises $1.28M$ `train` images and $50K$ `val` images across 1000 classes. HRVMamba is trained using the Swin Transformer (Liu et al., 2021) training framework on $8 \times 80$GB A100 GPUs.

**Results.** Table 6 compares HRVMamba with several representative CNN, ViT, and SSM methods. HRVMamba demonstrates competitive performance across isotropic architectures (Touvron et al., 2021; Li et al., 2024), hierarchical architectures (Liu et al., 2021; 2024b), and high-resolution architectures (Yuan et al., 2021). Specifically, HRVMamba-B achieves a Top-1 accuracy of 83.7, using only 30% of the FLOPs of VideoMamba-M, which achieves 83.8 in Top-1 accuracy. Although MambaVision-B achieves a higher overall accuracy of 84.2, it employs the more advanced LAMB optimizer, which is not used by the other models. Additionally, MambaVision-B is trained on 32 A100 GPUs, whereas our model utilizes only 8 A100 GPUs and does not rely on advanced training techniques like distillation (Shaker et al., 2024). Importantly, as shown in Table 3, MambaVision-B underperforms in dense prediction tasks, scoring 73.4 AP compared to 76.4 AP for HRVMamba-B. Notably, **HRVMamba achieves the best performance among high-resolution architectures**, with HRVMamba-S showing a 0.6-point improvement over HRFormer-S, and HRVMamba-B outperforming HRFormer-B by 0.5 points with comparable FLOPs.

Table 7: **Ablation Experiments Results on COCO `val` set.** All models are not pretrained on the ImageNet. The HRVMamba-S in Table 2 is the basic architecture setting. The input size is $256 \times 192$. † denotes their is only $3 \times 3$ depthwise convolution in MultiDW Block.

| SS2D Block | DSS2D Block | MultiDW Block | MutiDW in FFN | AP | AR |
|:---:|:---:|:---:|:---:|:---:|:---:|
| ✔ | ✗ | ✗ | ✗ | 73.5 | 78.8 |
| ✗ | ✔ | ✗ | ✗ | 73.9 | 79.2 |
| ✗ | ✔ | ✗ | ✔ | 73.5 | 79.1 |
| ✗ | ✔ | ✔ | ✗ | **74.2** | **79.5** |
| ✗ | ✔ | ✔† | ✗ | 73.9 | 79.2 |

## 5.4 Ablation Experiments

**Multi-resolution Parallel architecture.** The results in Table 3 demonstrate that HRVMamba, utilizing the Multi-resolution Parallel architecture, achieved SOTA performance in pose estimation. In particular, comparisons with other SOTA SSM models such as VMamba (Liu et al., 2024b), VMamba-B (Liu et al., 2024b), MambaVision-B (Hatamizadeh & Kautz, 2024), and GroupMamba-B (Shaker et al., 2024) highlight the advantages of the Multi-resolution Parallel architecture for dense prediction tasks.

**2D-Selective-Scan with Deformable Convolution.** As shown in Table 7, the DSS2D block improves AP by 0.4 points compared to the SS2D block (line 2 vs. line 1), demonstrating that incorporating DCN enhances Mamba's spatial feature extraction. Specifically, as shown in Fig. 1, DSS2D focuses on high-level features related to the query patch in the early stage (S2), while SS2D targets low-level edge features. In the later stage (S3), DSS2D highlights human-related details, whereas SS2D tends to capture irrelevant background information. We think DCNv4 enhances the features of high-level semantic relations between patches, allowing them to influence each other despite long-range decay (long-range forgetting issue).

**Multi-scale Depthwise Block.** HRFormer introduces depthwise convolution in the FFN to enhance the model's inductive bias. However, our experimental results in Table 7 show that embedding the MultiDW Block into the FFN (line 3 vs. line 2) can even degrade the performance gains brought by DCN. In contrast, when the MultiDW Block is used as a standalone module, as shown in Fig. 3, it improves the AP to 74.2. Yet, replacing the multi-scale convolutional kernels with $3 \times 3$ convolution does not result in any performance improvement (line 5 vs. line 3). This indicates that the multi-scale convolutional kernels are effective in capturing local features at various scales, thus strengthening the inductive bias for features.

## 6 Future Work

Our experimental results show that pretraining on ImageNet improves the performance of the smaller model, HRVMamba-S, in pose estimation. However, for the larger model, HRVMamba-B, it can even lead to a performance drop. This suggests that the pretraining strategy for SSM models might differ from that of CNNs and ViTs due to the unique mechanisms of SSM. There is significant room for further research into the pretraining strategy for HRVMamba. Exploring alternative pretraining approaches may potentially enhance HRVMamba's performance.

## 7 Conclusion

Visual Mamba's performance on dense prediction tasks faces challenges such as insufficient inductive bias, long-range forgetting, and low-resolution output representations. To address these issues, we introduce the Dynamic Visual State Space (DVSS) block, which employs multi-scale convolutional kernels to enhance inductive bias and utilizes deformable convolutions to mitigate long-range forgetting. By leveraging the multi-resolution parallel design from HRNet, we present the High-Resolution Visual State Space Model (HRVMamba), which maintains high-resolution representations throughout the network, ensuring effective multi-scale feature learning. Extensive experiments demonstrate that HRVMamba achieves competitive results across various dense prediction benchmarks compared to CNNs, ViTs, and SSMs.

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
