# OpenReview forum: "HRVMamba: High-Resolution Visual State Space Model for Dense Prediction"
_ICLR.cc/2025/Conference — ICLR 2025 Conference Withdrawn Submission_

### Official Review · Reviewer_YPDX · 2024-10-27

**Soundness:** 2
**Presentation:** 2
**Contribution:** 1
**Rating:** 3
**Confidence:** 4

**Summary:**

This paper summarizes the current challenges encountered when applying vision mamba for dense prediction tasks, including insufficient inductive bias, long-range forgetting, and low-resolution output representation. Subsequently, the authors propose corresponding solutions, including multi-scale convolution kernels, deformable convolution, and the HRNet framework to alleviate these issues.

**Strengths:**

This paper first introduces Vision Mamba to a multi-resolution branch architecture. Additionally, several techniques are introduced to alleviate the limitations of Vision Mamba, including deformable convolution and multi-kernel size convolution. These techniques provide improvements over the vanilla Vision Mamba baseline. The proposed methods are easy to follow.

**Weaknesses:**

1. This paper feels more like a technical report instead of an academic paper. The challenges of Vision Mamba models are obtained from previous works, and the solutions are also based on current techniques. Besides, some observations and methods are not properly cited. For example, the problem where multi-way scanning approach disrupts 2D spatial dependencies has been proposed in MambaIR[1], and the solution of Multi-scale Depthwise block has also been introduced in SMT[2]. Given these considerations, this paper lacks introducing any new insights or techniques to the community.

2. The experiments are not valid enough. For example, for the semantic segmentation, only results on Cityscapes and PASCAL Context are reported. The results on the widely used ADE20K are missing. The authors may consider reporting results with the same framework like Uppernet on ADE20K and compare with publicly available results like VMamba. Complementing these results and comparing with publicly available results will make this paper more solid.

3. Some experimental data lack further explanation. In Table 7, the authors report the performance on the COCO val set. As the COCO dataset has many benchmarks, this "val" set is ambiguous. If it refers to the pose estimation, the best results here are 74.2, which is not coherent with the results in Table 3. Additional explanation for the setting differences is needed. Besides, there are some very weird data such as the one in L412. The authors mentioned the poor performance of LocalVMamba on the PASCAL-Context dataset, but no further explanation for the reason is provided.

4. The detailed efficiency comparisons are missing, including the inference speed and memory cost on different datasets.

5. Some minor errors: In L213, Fig.3 refers to Fig.2 by mistake. Besides, the contents from L445 to L450 are not appropriate for the ablation section.

6. This work does not give any discussion about the limitations.

Reference:

[1]. Guo, Hang, et al. "MambaIR: A Simple Baseline for Image Restoration with State-Space Model." arXiv e-prints (2024): arXiv-2402.

[2]. Lin, Weifeng, et al. "Scale-aware modulation meet transformer." Proceedings of the IEEE/CVF International Conference on Computer Vision. 2023.

**Questions:**

Why does the performance of HRFormer-B on the segmentation datasets significantly lag behind the reported results? For example, HRFormer-B + OCR achieves a mIoU of 82.6 on Cityscapes and 58.5 on PASCAL-Context datasets, respectively. However, the performance drops to 77.3 and 42.6 in Table 5, Lines 349-350.

---

### Official Review · Reviewer_M2XD · 2024-11-04

**Soundness:** 3
**Presentation:** 3
**Contribution:** 2
**Rating:** 5
**Confidence:** 5

**Summary:**

The paper introduces HRVMamba, a High-Resolution Visual State Space Model designed for dense prediction tasks such as human pose estimation and semantic segmentation. HRVMamba addresses the limitations of previous Mamba models by incorporating a Dynamic Visual State Space (DVSS) block, which uses multi-scale convolutional kernels to enhance inductive bias and deformable convolutions to mitigate long-range forgetting. The model is based on a multi-resolution parallel design, preserving high-resolution representations throughout the network to facilitate effective multi-scale feature learning. Extensive experiments demonstrate HRVMamba's competitive performance against existing CNN, ViT, and SSM benchmark models on various dense prediction tasks.

**Strengths:**

1. The paper presents the Dynamic Visual State Space (DVSS) block, which combines multi-scale convolutional kernels and deformable convolutions.
2. This paper proposes the High-Resolution Visual State Space Model (HRVMamba) based on the DVSS block and the architecture of HRNet.
3. The proposed HRVMamba obtains improvements compared to previous approaches on several dense prediction masks

**Weaknesses:**

1. This paper's novelty is limited. It incorporates the VMamba block[1] into the HRNet architecture[2], including DCNv4, making it lack sufficient novelty and insights. In addition, similar ideas have been explored in HRFormer[3,4]. The novelty of this paper is below the threshold for ICLR publication.
2. The motivations for using DCNv4 and DVSS block in this paper are unclear. For example, this paper lacks a specific analysis of the long-range forgetting problem of Vmamba in vision and why DCNv4 can solve such a long-range forgetting problem of mamba. Whether from the theoretical or experimental analysis perspective, the authors need to provide exact evidence to present.
3. Lack of comparisons with recent vision mamba works, such as MambaVision[5].
4. In Fig.1, why are there neat blocks in the activation map? How can this be explained? Is it related to the scans in different directions of VMamba? Image activation is usually continuous, and I'm confused about it.
5. How do the feature maps of different resolutions fuse, downsample, or upsample?
6. How about the inference latency of the proposed HRVMamba, which includes VMamba blocks, multi-scale depthwise convolution blocks, and DCNv4 blocks?
7. In Tab.7, adding a 3x3 convolution has no effect. However, adding larger depth-wise convolutions, such as 5x5, 7x7, or 9x9, improves a little (0.3 AP), but this also introduces many additional parameters. It's unclear here whether the effect comes from extra parameters, larger convolutions, larger receptive fields, or multi-scale convolutions.
8. The performances of baseline methods (such as HRFormer) on Cityscapes and PASCAL Context are too low, which are far inconsistent with the original paper[3], for example, HRFormer-B obtains 82.6 mIoU (Cityscapes) and 58.5 mIoU (PASCAL Context) while achieving 77.3 mIoU (Cityscapes) and 42.6 mIoU (PASCAL Context) in this paper. I think a fair comparison is very important. However, the results of the current comparison methods are obviously much lower than those of the original methods.

References\
[1] Liu et al. VMamba: Visual State Space Model. NeurIPS 2024.\
[2] Wang et al. Deep High-Resolution Representation Learning for Visual Recognition. TPAMI 2020.\
[3] Yuan et al. HRFormer: High-Resolution Transformer for Dense Prediction. NeurIPS 2021.\
[4] Gu et al. Multi-Scale High-Resolution Vision Transformer for Semantic Segmentation. CVPR 2022.\
[5] Hatamizadeh et al. MambaVision: A Hybrid Mamba-Transformer Vision Backbone. NeurIPS 2024.

**Questions:**

1. The proposed HRVMamba add a multi-scale DW block and I'm concerned how about the performance of dropping the FFN block.

---

### Official Review · Reviewer_YccU · 2024-11-04

**Soundness:** 2
**Presentation:** 3
**Contribution:** 2
**Rating:** 5
**Confidence:** 5

**Summary:**

The paper introduces HRVMamba, a high-resolution visual state space model tailored for dense prediction tasks. It builds on the Mamba framework, a hardware-efficient state space model (SSM) known for linear computational complexity. The authors highlight limitations in previous visual Mamba models—namely, insufficient inductive bias, long-range forgetting, and low-resolution outputs. To overcome these, HRVMamba incorporates the Dynamic Visual State Space (DVSS) block, combining multi-scale convolutional kernels and deformable convolution for enhanced local and long-range feature extraction. The HRVMamba model employs a multi-resolution parallel structure inspired by HRNet, preserving high-resolution representations and facilitating multi-scale feature learning.

**Strengths:**

- Empirical results indicate that HRVMamba outperforms contemporary CNNs, ViTs, and SSMs on benchmarks, delivering competitive performance with fewer computational resources.
- The figures in the paper are clean and aesthetically pleasing, which enhances readability.

**Weaknesses:**

- Limited Novelty: The HRVMamba model mainly combines existing methods, including the VSS block, DWConv, DCN, and the HRNet architecture. This integration-based approach may not meet ICLR’s high standards for innovation.
- Concern for `Limitation 1`: While the paper addresses the lack of 2D inductive bias in previous visual Mamba models, it raises concerns about whether introducing such bias could restrict the **scaling ability** of HRVMamba. Vision Transformers (ViTs) have demonstrated that reduced inductive bias can facilitate better scaling, so incorporating strong inductive biases might limit HRVMamba's scalability and performance on larger-scale models.
- Concern for `Limitation 2`: The paper uses Deformable Convolutions (DCN) to mitigate the long-range forgetting issue observed in previous visual Mamba models. However, there is a concern about whether DCN can effectively address this problem as the sequence length scales up. The efficacy of DCN for maintaining high-level feature relationships over significantly longer sequences remains uncertain, raising questions about its robustness as a scalable solution for long-range dependencies.
- The paper references preprints and arXiv versions of significant works, such as Mamba (COLM), Vision Mamba (ICML), and VMamba (NeurIPS). The authors should update these citations to their final published versions to reflect the current state of the literature.

**Questions:**

Please refer to the weakness part.

---

### Official Review · Reviewer_Y5sm · 2024-11-06

**Soundness:** 3
**Presentation:** 3
**Contribution:** 3
**Rating:** 5
**Confidence:** 4

**Summary:**

The paper proposes HRVMamba, a High-Resolution Visual State Space Model designed for dense prediction tasks, such as human pose estimation and semantic segmentation. The paper addresses limitations in existing Mamba-based models, particularly Mamba’s low-resolution output and challenges in retaining long-range dependencies. To overcome these issues, the authors introduce the Dynamic Visual State Space (DVSS) block, which leverages multi-scale and deformable convolutions to improve inductive bias and mitigate long-range forgetting. By integrating these innovations within a high-resolution, multi-resolution parallel structure, HRVMamba achieves competitive results across dense prediction benchmarks compared to CNN, ViT, and SSM models.

**Strengths:**

HRVMamba demonstrates competitive or superior performance on COCO, Cityscapes, and PASCAL-Context benchmarks, often with fewer parameters and reduced computational load compared to similar models.

**Weaknesses:**

Limited Novelty: In my view, this paper incorporates techniques from CNN networks, such as DCNv4 and multi-resolution structures (from FPN and HRNet), into the Mamba block to enhance network performance. I am somewhat skeptical about whether such an innovation alone is sufficient for a publication at ICLR.

**Questions:**

See the weakness above

---

### Note · Authors · 2025-01-06

**Comment:**

We decided to withdraw the paper and improve it.

**Withdrawal Confirmation:**

I have read and agree with the venue's withdrawal policy on behalf of myself and my co-authors.